



# COVID-19 lockdown induced changes in NO₂ levels across India observed by multi-satellite and surface observations

Akash Biswal[1,2], Vikas Singh[1*], Shweta Singh[1], Amit P. Kesarkar[1], Khaiwal Ravindra[3],

Ranjeet S. Sokhi[4], Martyn P. Chipperfield[5,6], Sandip S. Dhomse[5,6], Richard J. Pope[5,6], Tanbir

Singh[2], Suman Mor[2]

1. National Atmospheric Research Laboratory, Gadanki, AP, India

2. Department of Environment Studies, Panjab University, Chandigarh 160014, India

3. Department of Community Medicine and School of Public Health, Post Graduate Institute

of Medical Education and Research (PGIMER), Chandigarh 160012, India

4. Centre for Atmospheric and Climate Physics Research (CACP), University of Hertfordshire,

Hatfield, UK

5. School of Earth and Environment, University of Leeds, Leeds, UK

6. National Centre for Earth Observation, University of Leeds, Leeds, UK

*Correspondence to:* Vikas Singh (vikas@narl.gov.in)

**Abstract**

We have estimated the spatial changes in NO₂ levels over different regions of India during the

COVID-19 lockdown (25th March – 3rd May 2020) using the satellite-based tropospheric

column NO₂ observed by the Ozone Monitoring Instrument (OMI) and the Tropospheric

Monitoring Instrument (TROPOMI), as well as surface NO₂ concentrations obtained from the

Central Pollution Control Board (CPCB) monitoring network. A substantial reduction in NO₂

levels was observed across India during the lockdown compared to the same period during

previous business-as-usual years, except for some regions that were influenced by anomalous

fires in 2020. The reduction (negative change) over the urban agglomerations was substantial

(~20-40 %) and directly proportional to the urban size and population density. Rural regions

across India also experienced lower NO₂ values by ~15-25 %. Localised enhancement of NO₂

associated with isolated emission increase scattered across India, were also detected. Observed

percentage changes in satellite and surface observations were consistent across most regions

and cities, but the surface observations were subject to larger variability depending on their



proximity to the local emission sources. Observations also indicate $NO_2$ enhancements of up
to ~ 25 % during the lockdown associated with fire emissions over the north-east, and some
parts of central regions. In addition, the cities located near the large fire emission sources show
much smaller $NO_2$ reduction than other urban areas as the decrease at the surface was masked
by enhancement in NO2 due to the transport of the fire emissions.
**Keywords:** OMI, TROPOMI, CPCB, Emission reduction, Air quality, ISRO LULC

# 1    Introduction

Nitrogen oxides $NO_x$ ($NO+NO_2$) are one of the major air pollutants, as defined by various
national environmental agencies across the world, due to its adverse impact on human health
(e.g. Mills et al., 2015). Furthermore, tropospheric levels of $NO_x$ can affect tropospheric ozone
formation (Monks et al., 2015), contribute to the secondary aerosol formation (Lane et al.,
2008), acid deposition, and impact climatic cycles (Lin et al., 2015). The major anthropogenic
sources of $NO_x$ emissions include the combustion of fossil fuels in road transport, aviation,
shipping, industries, and thermal power plants (e.g. USEPA, 1999; Ghude et al., 2013; Hilboll
et al., 2017). Other sources include open biomass burning (OBB), mainly large-scale forest
fires (e.g. Hilboll et al., 2017), lightning (e.g. Solomon et al., 2007) and emissions from soil
(e.g. Ghude et al., 2010). $NO_x$ hotspots are often observed over thermal power plants, industries
and urban areas with large traffic volumes causing larger localised emissions (e.g. Prasad et
al., 2012; Hilboll et al., 2013; Ghude et al., 2013).
With growing scientific awareness of the adverse impacts of air pollution, the number of air
quality monitoring stations has expanded to over 10,000 across the globe (Venter et al., 2020).
Additionally, multiple missions including the Global Ozone Monitoring Instrument (GOME)
on ERS-2, the Scanning Imaging Absorption Spectrometer for Atmospheric Cartography
(SCIAMACHY, 2002-2012) on Envisat, the Ozone Monitoring Instrument (OMI, 2005-
present) on Aura, GOME-2 (2007-present) on MetOp and the TROPOspheric Monitoring
Instrument (TROPOMI, 2017-present) on Sentinel-5P (S5P) have monitored $NO_2$ pollution
from the space for over two decades. Surface sites typically measure $NO_2$ in concentration
quantities (e.g. µg m$^{-3}$), but satellite $NO_2$ measurements are retrieved as integrated vertical
columns (e.g. tropospheric vertical column density, $VCD_{trop}$). The latter is preferred to study
$NO_2$ trends and variabilities because of global spatial coverage, and spatio-temporal similarity
with ground-based measurements (Martin et al., 2006; Kramer et al., 2008; Weing et al., 2008;
Lamsal et al., 2010; Ghude et al., 2011). $NO_2$ has been reported to increase in south Asian



countries (Duncan et al., 2016; Hilboll et al., 2017; ul-Haq et al., 2017), decrease over Europe
(van der A  et al., 2008; Curier et al., 2014; Georgoulias et al., 2019) and the United States (
Russell et al., 2012; Lamsal et al., 2015). In the case of India, tropospheric $NO_2$ increased
during the 2000s (Mahajan et al., 2015; Hilboll et al., 2017), but since 2012 it has either
stabilized or even declined owing to the combined effect of economic slowdown and adaptation
of cleaner technology (Hilboll et al., 2017). However, thermal power plants, megacities, large
urban areas and industrial regions remain the $NO_2$ emission hotspots (Ghude et al., 2008, 2013;
Prasad et al., 2012; Hilboll et al., 2013; Duncan et al., 2016; Hilboll et al., 2017). Moreover,
despite the measures taken to control $NO_x$ emissions, urban areas often exceed national ambient
air quality standards in India (Sharma et al., 2013; Nori-Sarma et al., 2020; Hama et al., 2020),
and thus require a detailed scenario analysis.
The nationwide lockdown in various countries during March-May 2020 due to the outbreak of
COVID-19 reduced the traffic and industrial activities leading to a significant reduction of
$NO_2$. Studies using space-based and surface observations of $NO_2$ have reported reductions in
the range of ~30-60 % for China, South Korea, Malaysia, Western Europe, and the U.S.
(Bauwens et al., 2020; Kanniah et al., 2020; Muhammad et al., 2020; Tobías et al., 2020;
Dutheil et al., 2020; Liu et al., 2020; Huang and Sun 2020; Naeger and Murphy 2020; NASA,
2020), with the reductions observed strongly linked to the restrictions imposed on vehicular
movement. The lockdown in India was implemented in various phases starting on the 25[th]
March 2020 (MHA, 2020; Singh et al., 2020). The lockdown restrictions in the first two phases
(Phase 1: 25[th] March - 14[th] April 2020 and Phase 2: 15[th] April to -3[rd] May 2020) were the
strictest, during which all non-essential services and offices were closed and the movement of
the people was restricted, resulting in a large reduction in the anthropogenic emissions. The
restrictions were relaxed in a phased manner from the third phase onwards in less affected areas
by permitting activities and partial movement of people (MHA, 2020).
A decline in $NO_2$ levels over India during the lockdown has been reported from both surface
observations (Singh et al., 2020; Sharma et al., 2020; Mahato et al., 2020), as well as satellite
observations (ESA, 2020; Biswal et al., 2020; Siddiqui et al., 2020; Pathakoti et al., 2020).  A
detailed study by Singh et al. (2020) based on 134 sites across India reported a decline of ~30–
70 % in $NO_2$ with a larger reduction observed during peak morning traffic hours and late
evening hours. While Sharma et al. (2020) reported a lesser decrease (18 %) in $NO_2$ for selected
sites, Mahato et al., (2020) found a decrease of over 50 % in Delhi for the first phase of
lockdown which was also confirmed by Singh et al. (2020) for the extended period of analysis.


The satellite-based studies by Biswal et al. (2020) and Pathakoti et al. (2020) estimated the
change in $NO_2$ levels using OMI observations whereas Siddiqui et al. (2020) utilised
TROPOMI to compute the change over eight major urban centres of India. Biswal et al. (2020)
reported that average OMI $NO_2$ over India decreased by 12.7 %, 13.7 %, 15.9 %, and 6.1 %
during the subsequent weeks of the lockdown. Similarly, Pathakoti et al. (2020) reported a
decrease of 17 % in average OMI $NO_2$ over India as compared to the pre-lockdown period and
a decrease of 18 % against the previous 5-year average. Moreover, both the study reported a
larger reduction of over 50 % over Delhi. Similarly, Siddiqui et al. (2020) also reported an
average reduction of 46 % in the eight cities during the first lockdown phase with respect to
the pre-lockdown phase. While recent studies have utilized either only satellite observations or
only surface observations, this study goes further by adopting an integrated approach by
combining both measurement types to investigate $NO_2$ level changes over India in response to
the COVID-19 pandemic using OMI, TROPOMI and surface observations over different
regions. As both OMI and TROPOMI have similar local overpass times of approximately 13:30
(Penn and Holloway, 2020; van Geffen et al., 2020), diurnal influences on the retrievals of $NO_2$
for both instruments are similar. Moreover, as both instruments use similar retrieval schemes,
their $NO_2$ measurements should be comparable with a suitable degree of confidence (van
Geffen et al., 2020; Wang et al., 2020). We estimate the changes in the $NO_2$ levels over different
land-use categories and urban sizes. In addition to this, we investigate the spatial agreement
between population density and $NO_2$ spatial variability observed at the surface. A key benefit
of this study will be to understand and assess the impact of reduced anthropogenic activity on
$NO_2$ from the satellite and surface observations. This study thus provides an improved
understanding of the spatial variations of tropospheric $NO_2$ for future air quality management
in India.
## 2 Data and methodology
### 2.1 Data
Satellite observations of $VCD_{trop}$ $NO_2$ were obtained from OMI (2016-2020) and TROPOMI
(2019-2020). Surface $NO_2$ observations (2016-2020) at 139 sites across India were from the
Central Pollution Control Board (CPCB). The period from 25[th] March to 3[rd] May each year is
defined as the analysis period. Average $NO_2$ levels during the analysis period in 2020 and
previous years are referred to as lockdown (LDN) $NO_2$ and business as usual (BAU) $NO_2$,





respectively. The BAU years for OMI and CPCB are 2016-2019 whereas for TROPOMI the
BAU year is 2019 because of the unavailability of earlier observations.
$NO_2$ data were analysed for six geographical regions (north, Indo Gangetic Plain (IGP), north-
west, north-east, central and south) of India (supplementary Fig. S1). The $NO_2$ changes over
various land-use categories (i.e. urban, cropland and forestland) have been analysed using
spatially collocated land-use land cover (LULC) data (NRSC, 2012) and OMI and TROPOMI
observed $VCD_{trop}$ $NO_2$. Visible Infrared Imaging Radiometer Suite (VIIRS) fire count data was
used to study the fire anomalies during the LDN and other analysis periods.

### 2.1.1   OMI NO₂

OMI has a nadir footprint of approximately 13 km × 24 km measuring in the ultraviolet-visible
(UV-Vis) spectral range of 270-500 nm (Boersma et al., 2011). It uses differential optical
absorption spectroscopy (DOAS) to retrieve $VCD_{trop}$ (i.e. $VCD_{trop}$ is the difference between the
total and stratospheric slant columns divided by the tropospheric air mass factor; (Boersma et
al., 2004). Here, we use the OMI $NO_2$ 30 % Cloud-Screened Tropospheric Column L3 Global
Gridded (Version 3) at a 0.25º × 0.25º spatial grid from the NASA Goddard Earth Sciences
Data       and       Information       Services       Center       (GESDISC)       available       at
(https://giovanni.gsfc.nasa.gov/giovanni/). Details of the retrieval scheme and OMI standard
product (Version 3) are discussed by e.g. Celarier et al., (2008) and Krotkov et al., (2017).

### 2.1.2   TROPOMI NO₂

TROPOMI has a nadir-viewing spectral range of 270–500 nm (UV-Vis), 675–775 nm (near-
infrared, NIR) and 2305–2385 nm (short wave-infrared, SWIR). In the UV-Vis and NIR
wavelengths, TROPOMI has an unparalleled spatial footprint of 3.5 km × 7.0 km, along with
7 km × 7 km in the SWIR (Veefkind et al., 2012). Details of the TROPOMI scheme and data
are discussed by Eskes et al. (2019) and Van Geffen et al. (2019). The time-averaged $VCD_{trop}$
$NO_2$ over India for the analysis period was obtained at 10 km × 10 km resolution from the
Google       earth-engine       (https://developers.google.com/earth-
engine/datasets/catalog/COPERNICUS_S5P_OFFL_L3_NO2). The source data are filtered to
remove pixels with QA (Quality Assurance) values less than 75 % which removes cloud-
covered scenes, part of the scenes covered by snow/ice, errors and problematic retrievals (Eskes
et al., 2019).



### 2.1.3 Surface NO₂ concentration

The hourly averaged surface $NO_2$ concentration at 139 sites (Fig. S1) for 2016-2020 across India was acquired from the CPCB CAAQMS (Continuous Ambient Air Quality Monitoring Stations) portal (https://app.cpcbccr.com/ccr/#/caaqm-dashboard-all/caaqm-landing). The data was further quality controlled by removing the outliers, constant values, and sites having less than 60 % data during the analysis period. Details of the surface observations are explained in Singh et al. (2020).

### 2.1.4 Land use land cover data

The high-resolution (50 m × 50 m) LULC data mapped with level-III classification for 18 major categories (NRSC, 2012) was obtained from the BHUVAN geo-platform (https://bhuvan-app1.nrsc.gov.in/thematic/thematic/index.php) of the Indian Space Research Organisation (ISRO). To quantify the changes over urban, crop and forest areas, the OMI and TROPOMI $NO_2$ at urban grids (category 1), cropland (category 2 to 5) and forestland (category 7 to 10) were extracted for further analysis. In order to match the OMI and TROPOMI grid resolution with the Indian LULC, the dominant LULC was considered within the OMI and TROPOMI grid. Supplementary Fig. S2 shows the high-resolution LULC data used in this study for cropland, forestland, and urban areas separately. Urban areas were further divided into four sizes as 10-50 km², 50-100 km², 100-200 km² and greater than 200 km² to study the change in $NO_2$ with respect to the size of the urban agglomeration.

### 2.1.5 VIIRS fire counts

The VIIRS aboard the Suomi National Polar-orbiting Partnership (S-NPP) satellite provides daily global fire count at a 375 m × 375 m spatial resolution (Schroeder et al., 2014; Li et al., 2018). The fire count data over India during the analysis period from 2016 to 2020 was obtained from the FIRMS (Fire Information for Resource Management System) web portal (https://firms.modaps.eosdis.nasa.gov/download/). The fire count data was gridded at 10 km × 10 km for each year by summing of fire counts falling on each spatially overlapping grid. The burnt area was calculated from the fire counts by multiplying with the VIIRS grid size (Prosperi et al., 2020).

### 2.1.6 Population data

The gridded population density (people per hectare, pph) data for 2020 has been taken from Worldpop (2017). Worldpop estimates the population density at approximately 100 m × 100 m (near equator) by disaggregating census data for population mapping using random forest





estimation technique using remotely sensed and ancillary data. Details of the pollution mapping
methodology can be found in Stevens et al. (2015).
## 2.2  Analysis methodology
The change in the NO$_2$ levels for each analysis period has been calculated by subtracting the
BAU NO$_2$ from LDN NO$_2$. We calculate the percentage change ($D$) using the following
equation
$$D = \frac{(LDN - BAU)}{BAU} \times 100$$

The analysis was done over the whole of India as well as over the separate considered regions
and selected LULC categories using open-source Geographic Information System (QGIS).
# 3  Result and Discussion
## 3.1  Fire count anomalies during the lockdown
It is well known that meteorological factors (e.g. wind, temperature, radiation etc) can affect
the NO$_2$ concentration as well as biogenic emissions (Guenther et al., 2012). In the case of the
present study, recent work  (e.g. Singh et al., 2020; Navinya et al., 2020; Sharma et al., 2020)
has shown that meteorological conditions remained relatively consistent over recent years
during the dates of the lockdown period. Therefore, we assume that the changes observed
during the lockdown were due to the change in the emissions. Moreover, we have assumed no
change in biogenic emissions because of similar meteorological conditions during the
lockdown period. Long-term satellite-derived fire counts suggest that Indian fire activities
typically peak during March-May (Sahu et al., 2015), predominantly over the north, central
and north-east regions (Venkataraman et al., 2006; Ghude et al., 2013). However, the spatial
and temporal distribution of fire events is largely heterogeneous (Sahu et al., 2015) meaning
an abrupt increase or decrease in fire activity could have a significant impact on NO$_2$ levels
over anomalous regions during the lockdown.
An investigation of fire counts during the 2020 lockdown (LDN analysis period), when
compared with the corresponding 2016-2020 average, highlights a substantial decrease over
the eastern part of central India and an increase over the western part of central India and north-
east. In Fig. 1a widespread fire activity (counts of 10-50) is shown across India such as the
central region (Madhya Pradesh, Chhattisgarh, Odisha), parts of Andhra Pradesh, the Western
Ghats in Maharashtra and north-east region (Assam, Meghalaya, Tripura, Mizoram and





Manipur). The fire anomaly during the lockdown (Fig. 1b) shows positive fire counts (5-20)
over the north-east region, west of Madhya Pradesh in central India and scattered locations in
South India. The negative fire anomalies (-20 to -5) observed over the central region
(Chhattisgarh and Odisha) suggests a decrease in fire activity during the 2020 lockdown period.
To minimise the impact of fire emission in our analysis, we have considered the grids with zero
fire anomaly to assess the changes in $NO_2$ during the lockdown.

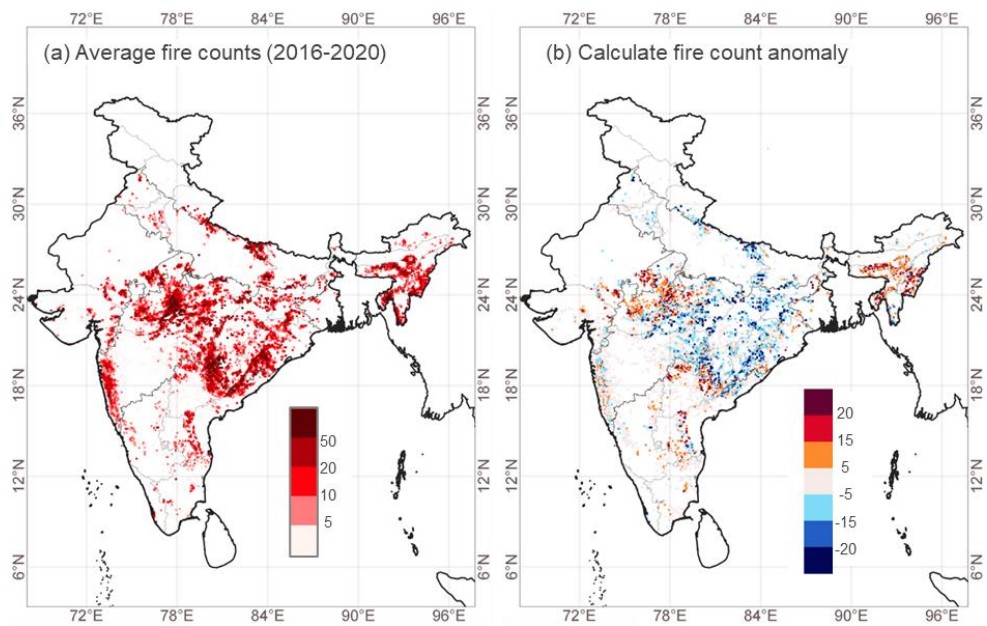

*Fig. 1 Spatial distribution of the 10 km × 10 km gridded VIIRS fire counts. (a) Average fire*
*counts during the analysis period (March 25th - May 3rd, 2016-2020). (b) Gridded fire*
*anomaly during the lockdown in 2020.*
**3.2    $VCD_{trop}$ $NO_2$ over India during lockdown period**
The spatial distribution of $VCD_{trop}$ $NO_2$ is largely determined by local emission sources;
therefore $NO_2$ hotspots are found over urban regions, thermal power plants and major industrial
corridors. For the Indian subcontinent, maximum $NO_2$ is observed during winter to pre-
monsoon (Dec-May) and minimum $NO_2$ during the monsoon (Jun-Sep). Region-specific peaks
such as the winter-time peak (Dec-Jan) in the IGP is associated with anthropogenic emissions,
or the summer-time peak (Mar-Apr) in central India and north-east India is associated with
enhanced biomass burning activities (Ghude et al., 2008; Ghude et al., 2013; Hilboll et al.,

236    2017).



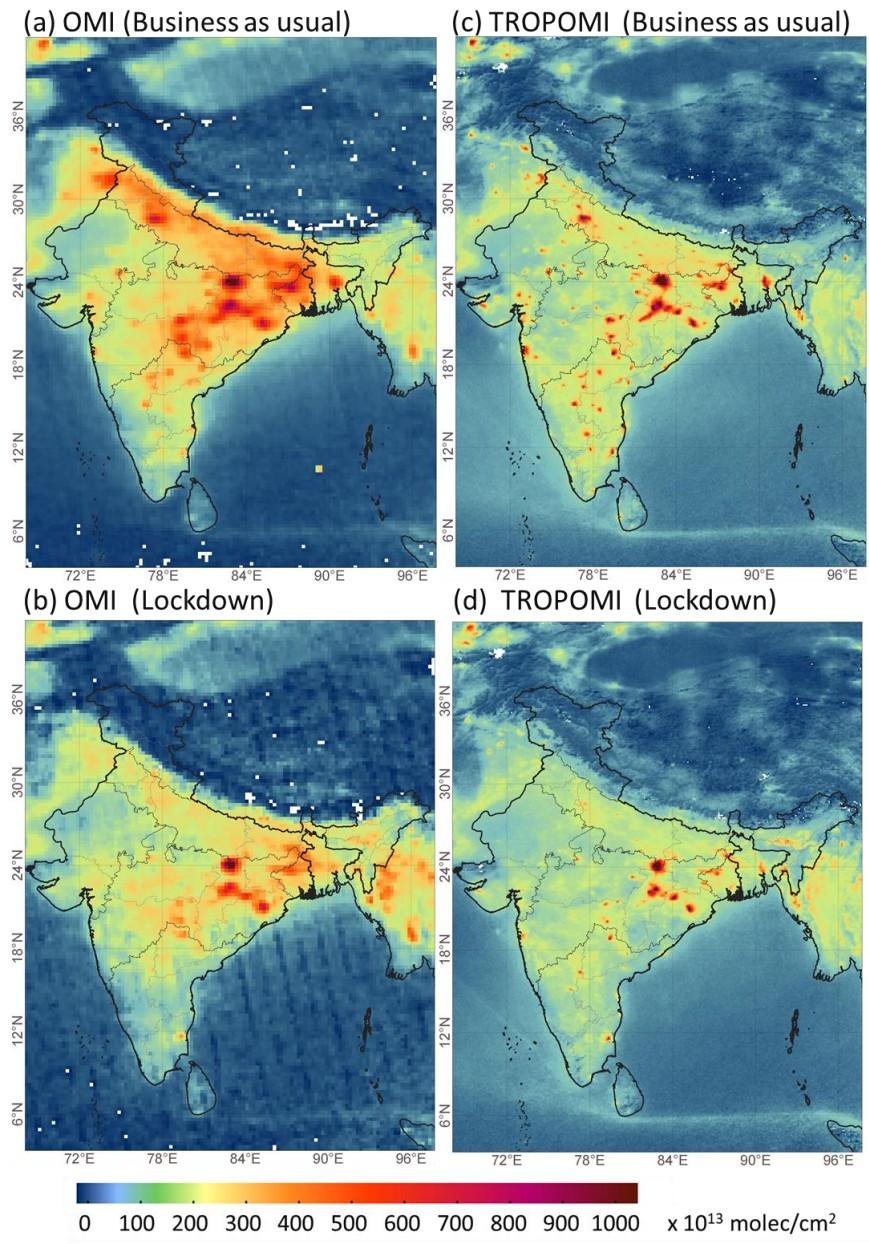


*Fig. 2 Spatial distribution of mean VCD$_{trop}$ NO$_2$ (molecules cm$^{-2}$) during the analysis period
(25$^{th}$ March - 3$^{rd}$ May) for (a) OMI NO$_2$ during business as usual (BAU, 2016-2019), (b) OMI
NO$_2$ during the lockdown (LDN, 2020), (c) TROPOMI NO$_2$ during BAU (2019) and, (d)
TROPOMI NO$_2$ during LDN (2020).*



We compare the LDN mean $VCD_{trop}$ $NO_2$ with the BAU mean for OMI and TROPOMI. The
spatial distribution of the BAU and LDN $VCD_{trop}$ $NO_2$ observed by OMI and TROPOMI is
shown in Fig. 2 (a-d). The mean $VCD_{trop}$ $NO_2$ from the two instruments show similar spatial
distributions during the analysis period for both LND and BAU. In BAU years, the $NO_2$
hotspots are seen over the large fossil-fuel-based thermal power plants ($\sim$1000 $\times10^{13}$ molecules
$cm^{-2}$), urban areas ($\sim$400-700 $\times10^{13}$ molecules $cm^{-2}$) and industrial areas. Scattered sources are
also present in western India, covering the industrial corridor of Gujarat and Mumbai, various
locations of south India, and densely populated areas (e.g. IGP). The spatial distribution shows
significant changes during the lockdown in 2020. The details of actual and percentage changes
are discussed in the subsequent sections.
**3.3   Changes observed by OMI and TROPOMI**
There is a substantial reduction in $VCD_{trop}$ $NO_2$ between the LDN and BAU (Fig. 3a & c). A
large reduction in the number of hotspots, mainly urban areas, is seen in both OMI and
TROPOMI observations. However, hotspots due to coal-based power plants remain during the
lockdown as electricity production was continued. Over the $NO_2$ hotspots, there has been an
absolute decrease of over 150 $\times10^{13}$ molecules $cm^{-2}$ ($\sim$250 $\times10^{13}$ molecules $cm^{-2}$ over
megacities) detected by both OMI and TROPOMI. Background $VCD_{trop}$ $NO_2$ has typically
reduced by approximately 30-100 $\times10^{13}$ molecules $cm^{-2}$ representing a percentage decrease of
30-50 % (OMI) and 20-30 % (TROPOMI) in rural regions (Fig. 3b & d). For urban regions,
both OMI and TROPOMI see a decrease of approximately 50 %, but reductions in smaller
urban areas are clearly noticeable in the TROPOMI data, given its better spatial resolution.
Both instruments observe an increase in $VCD_{trop}$ $NO_2$ in the north-eastern regions and moderate
enhancement over the western and central regions. These enhancements are linked with the
biomass burning activities during this period (Fig. 1).



*Fig. 3 (a, c) Absolute change and (b, d) percentage change in VCD$_{trop}$ NO$_2$ during the analysis*
*period for LDN year compared to BAU years as observed by OMI (left panels) and TROPOMI*
*(right panels).*


### 3.4    The change observed over different land use

Anthropogenic $NO_x$ emissions are typically more localised in urban and industrial centres, while biogenic sources (e.g. soil) are more important in rural regions. OBB activities peak in March-April (Sahu et al., 2015) and represent more sporadic sources. As the lockdown is expected to have reduced urban anthropogenic $NO_x$ sources (as shown in Fig. 3), it is important to assess the lockdown impact over the rural regions such as cropland and forestland as well.. In this section, we estimate the changes in $VCD_{trop}$ $NO_2$ over different land-types such as cropland, forestland, and urban areas (Fig. S2). To minimise the impact of OBB emissions in our analysis, we exclude grids with fire anomalies (Fig. 1) and those containing thermal power plants (Fig. S2d). However, absolute separation is not possible due to regional, and long-range transportation from nearby grids.

### 3.4.1    Changes over cropland and forestland

The changes in $VCD_{trop}$ $NO_2$ observed by OMI and TROPOMI over the cropland (Fig. S2a) in different regions of India are shown in Fig. 4a & 4b and Table S1. A decline in $VCD_{trop}$ $NO_2$ has been observed over croplands in all regions except for the north-east. A higher percentage decline was observed over IGP and south regions by both the satellites. While $VCD_{trop}$ $NO_2$ has decreased, prominent enhancements have been observed over the north-east and few grids in central and north-west regions. These enhancements can be attributed to the impact of nearby forest grids (Fig. 1). The observed changes over the forestland (Fig. 2.c) over different regions of India have been shown in Fig. 4(c, d) and Table S1. The average $VCD_{trop}$ $NO_2$ has declined over forestland in all the regions except for the north-east where $VCD_{trop}$ $NO_2$ was enhanced due to the positive fire anomaly (Fig. 1) during the analysis period. It can be noted that although we have taken the grids with zero fire anomaly, the effect of a nearby grid exhibiting positive fire anomaly cannot be ignored due to atmospheric dispersion and mixing. The inter-comparison of the changes observed by two satellites suggests that OMI data indicates a larger reduction in $VCD_{trop}$ $NO_2$ than TROPOMI in most of the regions.

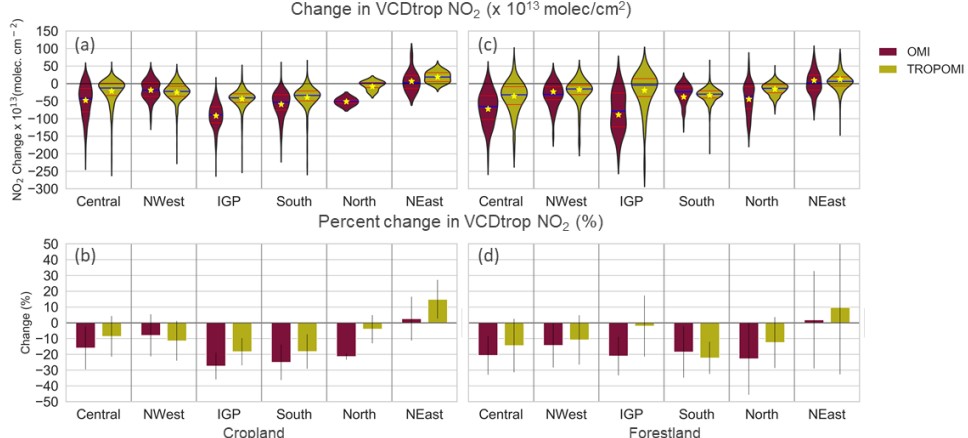

300

*Fig. 4 Observed change in VCD$_{trop}$ NO$_2$ between LDN and BAU from OMI and TROPOMI for different regions shown as (a) violin plot of the absolute change over cropland, (b) percentage change over cropland, (c) violin plot of the absolute change over forestland, and (d) percentage change over forestland. A violin plot is a combination of a box plot and a kernel density estimation (KDE) plot. KDE is a non-parametric way to estimate the probability density function (PDF). The red lines in the violin plot show the interquartile range; the blue line shows the median value; the yellow star shows the mean value. The vertical lines in the bar plot show the standard deviation The abbreviations NWest and NEast are for north-west and north-east regions, respectively.*

310

### 3.4.2 Changes over urban regions

Next, we analysed the changes in VCD$_{trop}$ NO$_2$ over the urban areas (Fig. S2b) in different regions of India. The calculated actual and percentage changes observed by OMI and TROPOMI are shown in Fig. 5 and in Table S1. The mean changes observed by OMI (in units $\times 10^{13}$ molecules cm$^{-2}$ (and %)) were -54 ± 48 (-22 ± 11 %) for the central region, -33 ± 26 (-14 ± 11 %) for the north-west, -110 ± 44 (30 ± 10 %) for IGP, -55 ± 37 (-25 ± 13 %) for the south, -92 ± 37 (-28 ± 6 %) for the north and 3±28 (2 ± 16 %) for the north-east. Similarly, the mean changes observed by TROPOMI (in the same units) were -65 ± 63 (-22 ± 15 %) for the central region, -74 ± 56 (-26 ± 14 %) for the north-west, -68 ± 46 (-23 ± 13 %) for IGP, -67 ± 49 (-26 ± 11 %) for the south, -43 ± 17 (-23 ± 8 %) for the north and 20±19 (16 ± 15 %) for the north-east. The changes observed over urban areas are larger than those observed over the forest and croplands. In contrast to the cropland and forestland, TROPOMI observed a larger reduction in VCD$_{trop}$ NO$_2$ than OMI in most of the regions. Densely populated IGP with the largest urban agglomeration shows the maximum change in VCD$_{trop}$ NO$_2$ followed by the central and north-west regions. The VCD$_{trop}$ NO$_2$ over the urban areas in the north-east region


is likely to be influenced by the nearby forest fires through atmospheric dispersion and mixing
resulting in the enhancement of $VCD_{trop}$ $NO_2$ over the urban grids.

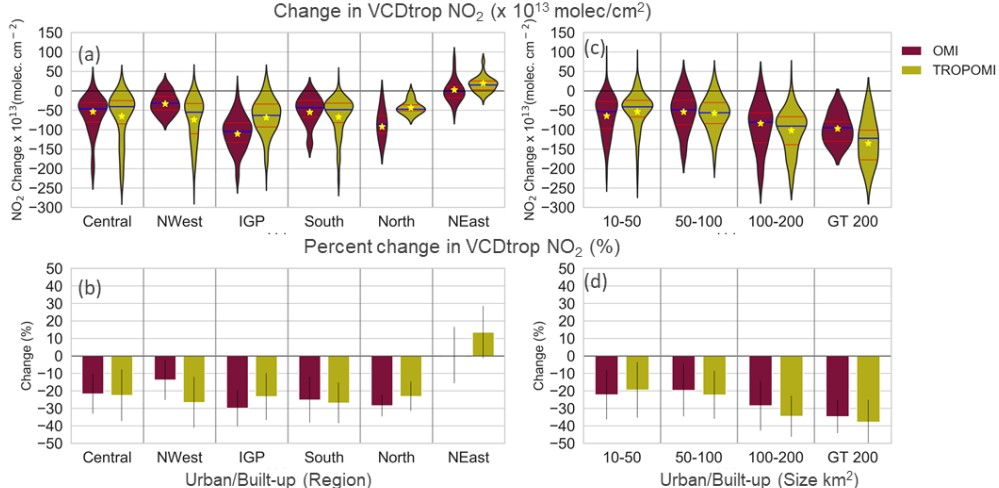


*Fig. 5 Observed change in $VCD_{trop}$ $NO_2$ between LDN and BAU from OMI and TROPOMI for*
*different regions shown as (a) Violin plot of the absolute change over urban areas, (b)*
*percentage change over the urban area, (c) violin plot of the observed change over different*
*sized urban areas, and (d) percentage change over different sized urban areas.*
We have also analysed the change in the $VCD_{trop}$ $NO_2$ over urban areas of different sizes. We
have taken the urban areas of sizes more than 10 $km^2$ and grouped them into four bins of size
10-50 $km^2$, 50-100 $km^2$, 100-200 $km^2$, and greater than 200 $km^2$. We then calculate the changes
observed for all the cities filling into the respective bins. Fig. 5 (c & d) show the absolute and
percentage change in $VCD_{trop}$ $NO_2$, as observed by OMI and TROPOMI, respectively. A
significant reduction of 50-150 $\times 10^{13}$ molecules $cm^{-2}$ (20-40 %) was observed over the urban
area of different sizes. The actual reduction in $VCD_{trop}$ $NO_2$ is greater for the larger urban area
with peak reductions for the urban area bin (> 200 $km^2$) for both OMI and TROPOMI.


**3.5    Inter-comparison of changes observed by OMI, TROPOMI and surface**
**observation**

Fig. 6 (a,b) shows the relationship of OMI and TROPOMI $NO_2$ with surface $NO_2$ for the BAU
and LDN periods, respectively. During BAU, there are reasonable positive correlations
between the satellite instruments and the surface sites (OMI: 0.44, TROPOMI: 0.47). In LDN,





these correlations drop to 0.3 and 0.23, respectively, potentially linked with the primary
reduction in urban $NO_2$ levels. We also determined the correlation between satellite and
surface-observed changes during the lockdown (Fig. 6c), finding values of 0.23 (OMI) and
0.36 (TROPOMI). This indicates that the lockdown $NO_2$ reductions appear to be present in
both measurement types, providing us with confidence in the observed changes detected in this
study.

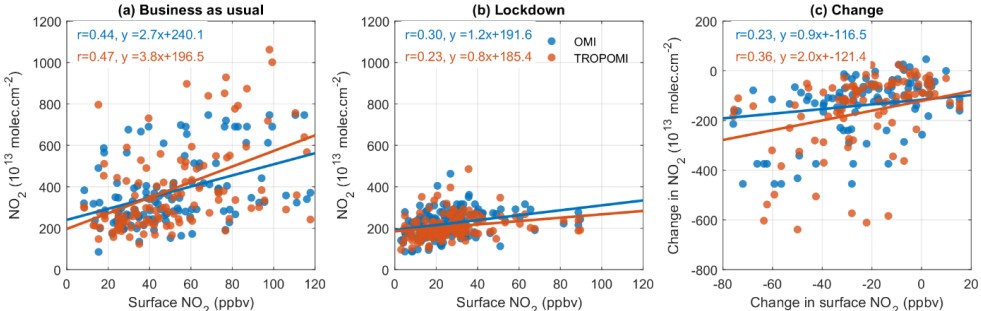

*Fig. 6 Scatterplots between surface and satellite observed $NO_2$ for (a) business as usual*
*(BAU) and (b) lockdown (LDN). Panel (c) shows a scatterplot of observed absolute change*
*(LDN-BAU) in surface and satellite $NO_2$.*

The LND $NO_2$ percentage change, observed by surface and spatially co-located satellite
measurements is shown in Figure 7 for various Indian regions. For this comparison, the number
of available CPCB surface monitoring stations were 17, 15, 81, 25, and 1 for central, north-
west, IGP, south and north-east regions (north region data not available), respectively. Most of
the CPCB stations are in urban areas, so our results reflect changes in predominantly urban-
sourced $NO_2$. At all surface sites in all regions, there was a percentage reduction greater than
20 % (Fig. 7). Satellite observations show a similar trend except for the north-east region where
enhancements are due to forest fires. Both OMI and TROPMI observed the highest reduction
(~50 %) over IGP. A smaller average reduction of ~20 % over central India might be due to
the aggregate effect of power plants, forest fires and prevalent biomass burning activities
during this season. While the effect of forest fires can be observed in the column $NO_2$, its effect
on the surface $NO_2$ is minimal. For the central, IGP and south regions, the mean percentage
change observed by the surface monitoring station is comparable to that observed by the
satellites.






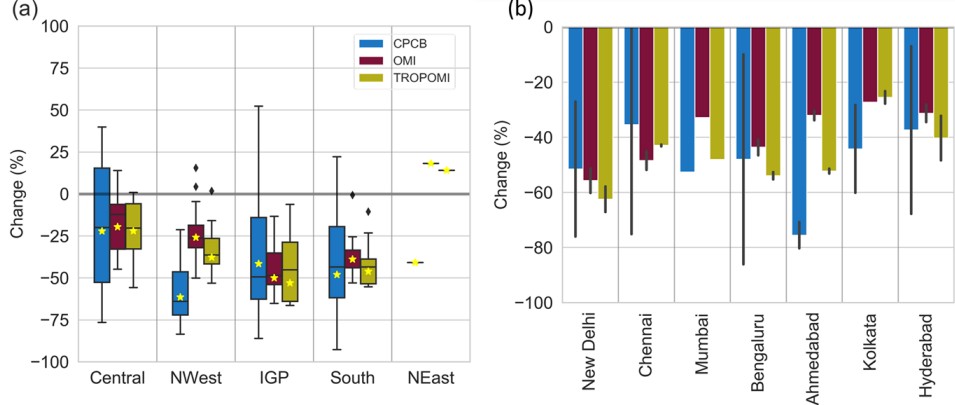


*Fig. 7 (a) Boxplot showing the percentage change between LDN and BAU in NO₂ levels*
*observed by ground and satellite measurements at CPCB monitoring locations in different*
*regions. (b) Barchart showing the percentage change in NO₂ levels observed at megacities in*
*India for the same measurements as panel (a). The vertical line in the barchart is the standard*
*deviation.*

We have intercompared the percentage change in $NO_2$ observed at the surface and by satellite

over the major Indian cities (i.e. New Delhi, Chennai, Mumbai, Bangalore, Ahmedabad,

Kolkata, and Hyderabad, Fig. 7b). A significant reduction in the range of ~25-75 % is observed,

consistent in all observational sources used in this study. A similar reduction observed by the

satellites over the cities in other parts of the world has been reported (Tobías et al., 2020;

Naeger and Murphy, 2020; Kanniah et al., 2020; Huang and Sun, 2020). The satellites observe

the largest reduction over Delhi and smallest over Kolkata. While the observed decline is

comparable for cities, Ahmedabad and Kolkata showed smaller declines than observed by

ground measurements. Also, the reduction observed at the surface has a larger spatial

variability than the one observed from the space. This is potentially linked to the influence of

the local emissions which could not be detected by the space-based instruments because of

relatively large satellite footprints. The results of percentage change observed by OMI are

consistent with the change reported by Pathakoti et al. (2020), although Siddiqui et al. (2020)

reported a higher decline of $NO_2$ using TROPOMI. This is because we computed the changes

between lockdown and BAU during the same period of the year whereas Siddiqui et al. (2020)

estimated the changes between the pre-lockdown $NO_2$ and the lockdown $NO_2$ which includes





the seasonal component of NO$_2$. We have also analysed the changes in VCD$_{trop}$ NO$_2$ observed
by both OMI and TROPOMI for the other major cities (Guttikunda et al., 2019), as shown in
Fig. S3. A reduction of over 20 % was observed in most of the cities except for a few in the
north-east and central India. Cities showing enhancement or smaller reductions reflect the
enhanced fire activities in the north-east and central Indian regions. TROPOMI can capture the
reduction over the cities near the fire-prone areas (e.g. Indore and Bhopal) because of its higher
spatial resolution.

**3.6   Correlation of tropospheric columnar NO$_2$ with the population density**

In this section, we examine the VCD$_{trop}$ NO$_2$ and population relationship for India except where
fire anomalies or large thermal power plants existed. The scatter density plots between VCD$_{trop}$
NO$_2$ and population density for the BAU and LDN analysis period are shown in Fig. 8 for OMI
and TROPOMI. The data were log-transformed to establish the log-log relationship as both
data sets are not normally distributed. As the observed changes had negative values, this log
transformation was obtained by adding a constant value which was later subtracted when
plotting to display the corresponding NO$_2$ values. Both OMI and TROPOMI NO$_2$ show a
similar relationship with the population density with correlations of ~0.7 during the LDN and
BAU periods, suggesting a strong dependence upon the population (i.e. anthropogenic
emissions). The slopes of the lines in Fig. 8 (a,b,d,e) show that VCD$_{trop}$ NO$_2$ follows a power-
law scaling with population density (Lamsal et al., 2013). During BAU, the VCD$_{trop}$ NO$_2$
observed over a grid increased by factors of 2.2 and 1.73 for OMI and TROPOMI, respectively,
with a ten-fold increase in the population density. The rate of increase of the VCD$_{trop}$ NO$_2$
during LDN was 2.0 and 1.58 times for OMI and TROPOMI, respectively, which was lower
than BAU. The correlation during the LDN period was marginally lower than the BAU period.
This could be due to a larger reduction in the NO$_2$ levels in the densely populated grids. The
changes observed in the VCD$_{trop}$ NO$_2$ during the LDN (Fig. 8c & f) were negatively correlated
(i.e. reduction was positively correlated) with the population density. The linear relation
suggests an increase in the reduction with an increase in the population density, however, some
grids exhibit enhancements in VCD$_{trop}$ NO$_2$ due to the local emissions.

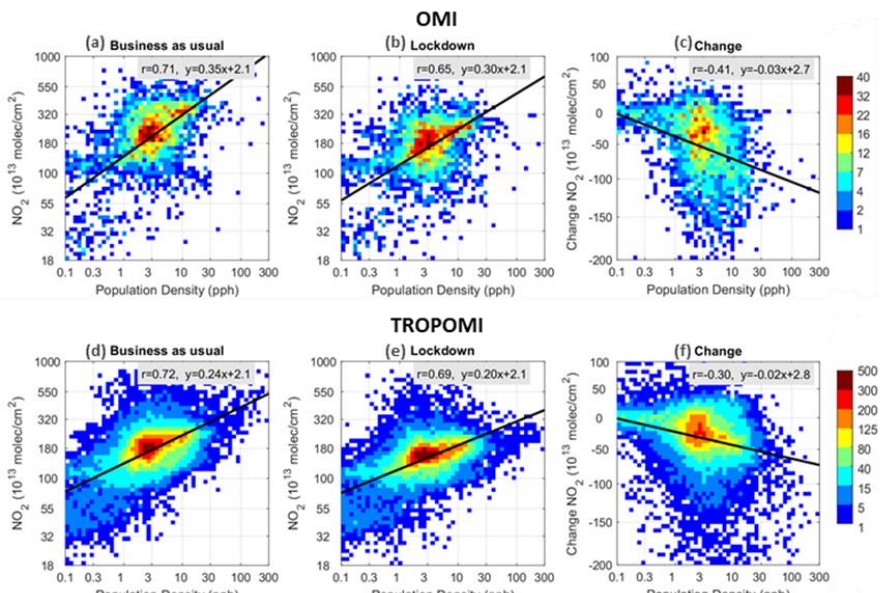


*Fig. 8. Scatter density plot between the VCD$_{trop}$ NO$_2$ ($\times 10^{13}$ molecules cm$^{-2}$) and population density (pph) for the analysis period in different years. (a) Business as usual (BAU, 2016-2019) observed by OMI; (b) lockdown (LDN, 2020) observed by OMI; (c) changes (LDN-BAU) observed by OMI; (d) BAU (2019) observed by TROPOMI; (e) LDN (2020) observed by TROPOMI; (f) LND-BAU changes observed by TROPOMI. The x and y axes are in log10 scale. The slope of the line is also shown in the log10 scale.*

## 4 Conclusions and discussion

The changes in NO$_2$ levels over India during the COVID-19 lockdown (25[th] March-3[rd] May 2020) have been studied using satellite-based VCD$_{trop}$ NO$_2$ observed by OMI and TROPOMI, and surface NO$_2$ concentrations obtained from CPCB. The changes between lockdown (LDN) and the same period during business as usual (BAU) years have been estimated over different land-use categories (e.g. urban, cropland, and forestland) across six geographical regions of India. Also, the changes observed from space and at the surface have been inter-compared and the correlation with the population density has been studied.

Overall, a significant reduction in NO$_2$ levels of up to ~70 % was observed over India during the lockdown as compared to the same period during BAU. The usual prominent NO$_2$ hotspots observed by OMI and TROPOMI over urban agglomerations during BAU were barely noticeable during the lockdown. However, the coal-based thermal power plants continued to be major NO$_2$ hotspots during the lockdown. Some of the largest reductions in NO$_2$ were observed to be over the urban areas of the IGP region. The reduction observed for urban



agglomerations was over $150 \times 10^{13}$ molecules cm$^{-2}$ (~30 %), and even more for megacities
showing a reduction of around $250 \times 10^{13}$ molecules cm$^{-2}$ (50 %). The reduction observed over
the urban areas was linked with reduced traffic emissions due to travel restrictions for COVID
containment. The reduction was also observed over rural regions. Average declines of $NO_2$ in
the ranges of 14-30 %, 8-28 % and 10-24 % were observed by OMI and 22-27 %, 6-18 % and
3-21 % were observed by TROPOMI over the urban, cropland and forestland, respectively, in
different regions of India. In contrast, an average enhancement over north-east India was
observed due to positive fire anomalies during the lockdown. Although we have considered the
grids with zero fire anomaly during the lockdown, the fire emissions can still contribute to the
enhancement of $NO_2$ levels over grids with no fire activity because of horizontal transport.
The observed changes in $VCD_{trop}$ $NO_2$ were found to be spatially positively correlated with
surface $NO_2$ concentrations indicating that the lockdown $NO_2$ changes appear to be present in
both measurement types. The TROPOMI $NO_2$ showed a better correlation with surface $NO_2$
and was more sensitive to the changes than the OMI because of the finer resolution. Therefore,
TROPOMI can provide a better estimate of $NO_2$ associated with fine-scale heterogeneous
emissions. Also, $VCD_{trop}$ $NO_2$ was found to exhibit a good correlation with the population
density, suggesting a strong dependence upon the population and hence the anthropogenic
emissions. The changes observed in the $VCD_{trop}$ $NO_2$ during the lockdown were negatively
correlated (i.e. reduction was positively correlated) with the population density suggesting a
larger reduction for the densely populated cities. However, the influence of local emissions can
be different in different cities.
The analysis presented in this work shows a significant change in $NO_2$ levels across India. The
observed reductions can be linked with the control measures taken to prevent the spread of the
COVID-19 that restricted the movement of the people resulting in a significant reduction in
anthropogenic emissions. As an important message to policymakers, this study indicates the
level of reduction in $NO_2$ that is possible if dramatic reductions in key emission sectors such
as road traffic, were incorporated into air quality management strategies.
**5    Data availability.**
The tropospheric columnar NO2 data for TROPOMI and OMI are available at Google earth-
engine (https://developers.google.com/earth-engine/) and NASA's Giovanni
(https://giovanni.gsfc.nasa.gov/giovanni/) respectively. Surface measured NO2 data across
India are available at CPCB site (https://app.cpcbccr.com/ccr/). VIIRS fire count data is



available at FIRMS web portal (https://firms.modaps.eosdis.nasa.gov/). India Population data used in this study is available at the https://www.worldpop.org/. The LULC data for India is available at the Bhuvan, (https://bhuvan.nrsc.gov.in/) Indian Geo-Platform of Indian Space Research Organisation.

## 6 Author contribution

**Akash Biswal and Vikas Singh:** Conceptualization, investigation, visualization, formal analysis, writing original draft, writing, reviewing and editing; **Shweta Singh:** Investigation, writing original draft, discussion, reviewing and editing, **Amit Kesarkar, Ravindra Khaiwal, Ranjeet Sokhi, Martyn Chipperfield, Sandip Dhomse, Richard Pope, Tanbir Singh, Suman Mor:** Investigation, discussion, reviewing and editing.

## 7 Declaration of competing interest

The authors declare that they have no known competing financial interests or personal relationships that could have appeared to influence the work reported in this paper.

## 8 Acknowledgments

The authors are thankful to the Director, National Atmospheric Research Laboratory (NARL, India), for encouragement to conduct this research and provide the necessary support. AB and SS greatly acknowledge the Ministry of Earth Sciences (MoES, India) for research fellowship. We acknowledge and thank Central Pollution Control Board (CPCB), Ministry of Environment, Forest and Climate Change (MoEFCC, India) for making available air quality data in public. We acknowledge Bhuvan, Indian Geo-Platform of Indian Space Research Organisation (ISRO), National Remote Sensing Centre (NRSC) for providing high-resolution LULC data. The authors gratefully acknowledge OMI and TROPOMI science teams for making OMI and TROPOMI data publicly available. We also acknowledge the NASA Giovanni and Google Earth Engine. We acknowledge support from the Air Pollution and Human Health for an Indian Megacity project PROMOTE funded by UK NERC and the Indian MOES, Grant reference number NE/P016391/1.





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
