# Peer review of "COVID-19 lockdown induced changes in NO₂ levels across India observed by multi-satellite and surface observations"

_Atmospheric Chemistry and Physics, 2020_

## Referee Comment (RC1) · Anonymous Referee #1 · 10 Dec 2020

The authors investigate the NO2 changes over India during COVID-19 lock-down period using both satellite and in-situ measurements. The authors investigated the differences between rural and urban areas. The contributions from natural sources are also considered. The manuscript is easy to follow and the primary conclusions are sound. I recommend publication after revisions.

General comments: 1. Section 3.1. The authors considered the grids with zero fire anomaly to assess the changes in NO2 during the lockdown. How about the grid cells surrounding big fires? I would suggest remove those grids from final analysis as well, since their NO2 patterns are very likely driven by fires. 2. Section 3.5. I would suggest

more investigation on the comparison between satellite and ground measurements. I'm not surprised by the low correlation between those two datasets in Figure 6. However, I don't see the reason why the correlation is even smaller during lock-down period. Please clarify this in the text. In addition, which datasets can represent the local NOx emission changes better? Does the difference indicate the uncertainty of one dataset? I suggest addressing those questions when performing the analysis. 3. Section 3.6. The authors remove grid cells with fire counts and power plants. How about other industrial plants? Will the grids with industrial plants bias the correlation between NO2 and population density? 4. Conclusion. "The reduction observed over the urban areas was linked with reduced traffic emissions due to travel restrictions for COVID containment." I would suggest a comparison with mobility data to support this conclusion. 5. simultaneous meteorology conditions. The authors mentioned that meteorology conditions constant during recent years by citing some references. In this way, the natural emissions are not the driver of the emission changes. Since this is the foundation of the whole analysis, I recommend a sub-session to clarify this point.

Specific comments: 1. Page 2, line 47. I suggest using the term of large to replace larger in the phase of larger localised emissions. 2. Page 2, line 59. The description of "spatio-temporal similarity with ground-based measurements" is confusing. Do the authors indicate the satellite and ground measurements share the similar spatial and temporal resolution? 3. Figure 4. I suggest adding a map to show the definition of the domain of Central, NWest, IGP and so on. It will be easier for readers to follow.

---

## Referee Comment (RC2) · Anonymous Referee #2 · 15 Dec 2020

Many papers have recently appeared in the literature, including papers on India, documenting reductions linked to COVID-19 measures. A large fraction of those papers made use of TROPOMI and OMI observations. Therefore I asked myself the question: what is new, and what have I learned? The authors write: "While recent studies have utilised either only satellite observations or only surface observations, this study goes further by adopting an integrated approach by combining both measurement types" Indeed, the paper contains interesting plots showing how surface and satellite analyses agree well, and document the relative reductions during LDN for the regions and cities in India. What I found also interesting is the analysis of the land use dependence and impact of changes in fire activity. The paper by Biswal et al. is a well written, easy

to read and clear paper, with good English. The paper contains an extended set of relevant references.

Because of the above I am in favour of publishing the paper after my comments have been taken into account.

Weather variability normally has a large impact on the BAU/LDN ratios. and can easily cause differences of the order of 20% in local differences between years. It is a bit of a pity that this aspect has not been explored by the authors. The paper is based on observations only, while models would need to be introduced to compensate for weather variability. Nevertheless, it would be good if the authors could provide a rough uncertainty estimate for the BAU/LDN relative changes due to the neglect of weather variability.

Substantial differences are found between the OMI and TROPOMI products: can those be understood? Newer satellite datasets are available for OMI. Why have those not been used?

It would be interesting to include also an analysis of the reductions of the major coal power plants, similar to the reductions observed for individual cities.

Due to transport of pollution, background values will be largely caused by downstream NO2 from the cities and power plants. This is not discussed well enough, and influences the land-use dependent analysis.

Detailed comments:

Intro, page 3/4: when referring to "decreases" I would always like to know with respect to what: is it compared to climatology, to concentrations before the lockdown, or in comparison to 2019? I suggest this information is provided in most cases (now it is given for only some references).

l110: "Moreover, as both instruments use similar retrieval schemes, their NO2 measurements should be comparable with a suitable degree of confidence?" I understand

that there are also major differences between these retrievals. These could be high-lighted more.

l147: The resolution of TROPOMI in 2020 is 3.5 x 5.5 km

l150: TROPOMI has a (2x) better resolution than 10x10 km. Why was this resolution chosen? 5x5 would be more appropriate.

sec 2.1.1, 2.1.2: I miss papers discussing the validation of the OMI and TROPOMI products.

sec 2.1.1, 2.1.2: The authors suggest that the retrieval approaches are comparable. However, there are also major differences. Compared to NASA v3 there are two newer retrieval products available: NASA v4 and the QA4ECV products. Have the authors considered using those?

Sec. 2.1.3: Is there a classification of these stations available (e.g. traffic, urban background etc..)? Please comment. Can stations be compared, and is a correlation with the satellite data meaningful?

l188: "Details of the pollution mapping". Do you mean "population mapping"?

Sec. 3.1. If I look at the satellite data it seems there is some impact of the fires, but the fire signal seems to be rather modest compared to the cities/power plants. For the fire anomalies I expect the differences to be even smaller. Why is it so important to exclude fire-influenced locations?

l222: "we have considered the grids with zero fire anomaly to assess the changes in NO2 during the lockdown." Plumes from the fires may be transported over long distances, so blocking fire pixels does not mean the fire signals have been removed.

l258: "Background VCDtrop". How is it determined if a pixel is "background"?

l265: "biomass burning activities" From figure 1 it is not so clear if there was an increase or decrease in fire activity in the West. Could you comment on this?

Sec 3.4: "The change observed over different land use" This title is a bit unclear, please reformulate

Sec 3.4.1: Land use changes occur over short distances. In contrast, the transport of (elevated) NO2 can happen over hundreds of kilometres. Therefore, the emissions from urban areas or coal plants will enhance concentrations over crop/forest grid cells, which blurs the interpretation. Also, differences in spatial resolution between OMI and TROPOMI may have similar effects. As such, there is a bit of a danger to over-interpret the results of e.g. Fig. 4.

Sec. 3.4.2: I would suggest not to repeat numbers from the table inside the text. Better to highlight a few.

l339: "The actual reduction in VCDtrop NO2 is greater for the larger urban area". But what is the reason? Is it related to different reductions in those cities or to differences in the relative contribution of the background not caused by emissions inside the city?

Sec 3.5, fig. 6: Without knowledge of the station classification or location relative to the sources these plots are not so meaningful. I would suggest to remove them, or put them in the supplement. Combining road-side and urban background stations in the plots will strongly reduce correlations.

Fig. 7: Ahmedabad is a clear outlier. Can the difference between surface and satellite observations be understood, for instance with information on the locations of the surface sites?

l 411: "adding a constant". This is not a logical thing to do in case of log-log plots, because the impact will be very non-linear. But the vertical scale in 8c is linear? Please re-write or remove the explanation.

l 415: It is not easy for me to judge from the figure panels if the relation follows a power law. And what does a factor 2.0 for OMI tell me?

---

## Author Comment (AC1) · 9 Feb 2021

Please see supplement for response to the referee #1

Please also note the supplement to this comment:
https://acp.copernicus.org/preprints/acp-2020-1023/acp-2020-1023-AC1-supplement.pdf

———————————————

---

## Author Comment (AC2) · 9 Feb 2021

Please see supplement for response to the referee #2

Please also note the supplement to this comment:
https://acp.copernicus.org/preprints/acp-2020-1023/acp-2020-1023-AC2-supplement.pdf
* * *

---

## Author Response (AR1)

Authors' responses to Referee #1. Reviewer's comments are in black text and authors' responses are in blue text.

**Anonymous Referee #1**

The authors investigate the NO2 changes over India during COVID-19 lock-down period using both satellite and in-situ measurements. The authors investigated the differences between rural and urban areas. The contributions from natural sources are also considered. The manuscript is easy to follow and the primary conclusions are sound. I recommend publication after revisions.

We thank you for recommending the publication of the article and appreciate the comments on the novelty of the study. Your constructive comments have helped us to improve the manuscript further. In the revised manuscript, the analysis has been updated using  $5 \text{ km} \times 5 \text{ km}$  TROPOMI VCDtrop NO2. We have also used ERA-5 data to investigate the changes in meteorological parameters (temperature, wind speed and boundary layer height) between BAU and lockdown. Section 3.8 linking the Google mobility change with NO2 change has been added. The limitations of the study have also been discussed in Section 3.9.

**General comments:**

1. Section 3.1. The authors considered the grids with zero fire anomaly to assess the changes in NO2 during the lockdown. How about the grid cells surrounding big fires? I would suggest remove those grids from final analysis as well, since their NO2 patterns are very likely driven by fires.

By considering the grids with zero fire anomaly, we excluded almost all the grids which have recorded fire activity during the analysis period. Upon further investigation, we find that a fire grid was surrounded by nearby fire grids in most of the cases (Figure 1 shown below) because the fires are likely to happen in fire-prone areas. Therefore, we mostly excluded the nearby grids covering big fires in our analysis. However, as the fire-plumes can be transported to long distances (longer than the nearby grids), the impact of long-range transport of forest fire plumes cannot be ignored over the areas with no fire activity. In the revised manuscript, we have included it as one of the limitations of our study in Section 3.9.

Figure 1. Zoomed map of fire activity over Central India. Blue circles indicate the clusters of fire activity

2. Section 3.5. I would suggest more investigation on the comparison between satellite and ground measurements. I'm not surprised by the low correlation between those two datasets in Figure 6. However, I don't see the reason why the correlation is even smaller during lockdown

period. Please clarify this in the text. In addition, which datasets can represent the local NOx emission changes better? Does the difference indicate the uncertainty of one dataset? I suggest addressing those questions when performing the analysis

The comparison between satellite data and ground-based measurements has its limitations and it is reasonable to raise a suspicion of the low correlation. The low correlation (0.45) between OMI and surface NO2 was also reported in Ghude et al. (2011). While they used data from a single site, our study includes multiple sites representing the satellite instruments' ability to capture the spatial heterogeneity. One of the reasons for the lower correlation can be the choice of surface station. Generally, urban background sites are preferred for this kind of analysis. However, the surface NO2 monitoring station type classification is not available for the CPCB sites; therefore, sites used in our analysis could be potentially impacted by traffic emissions resulting in lower correlation. Moreover, in-situ measurements are more sensitive to the local emission sources, whereas remotely sensed measurements provide values averaged over space as well as time. Therefore, in-situ measurements have larger variability than remotely sensed observations resulting in a low correlation.

Our analysis suggests that OMI and TROPOMI are sensitive to the emission changes at the surface because of positive correlation between the changes observed by space-based observations (OMI and TROPOMI) and the surface measurements. We find that TROPOMI has better sensitivity to changes than OMI because of a higher correlation.

Further, the reason for lower correlation during the lockdown can be linked to the lower NO2 levels (i.e., lower signals), resulting in a lower signal-to-noise ratio, therefore having larger uncertainty. We have updated this in the manuscript (now in Section 3.6)

3. Section 3.6. The authors remove grid cells with fire counts and power plants. How about other industrial plants? Will the grids with industrial plants bias the correlation between NO2 and population density?

India's industrial locations are often part of the urban agglomerates scattered around the city and are part of urban emissions. Therefore, we did not remove the industrial locations. To check for the bias, we have calculated the correlation between NO2 and population density after removing the data from industrial location and did not find a large difference in the correlation.

4. Conclusion. "The reduction observed over the urban areas was linked with reduced traffic emissions due to travel restrictions for COVID containment." I would suggest a comparison with mobility data to support this conclusion.

Thank you for the suggestion. In order to link the observed reduction in NO2 levels with the traffic emissions over the urban areas, we analyzed the Google mobility percentage reduction for three mobility categories: transit stations, workplace and residential, along with daily percentage change observed by OMI, TROPOMI and CPCB across urban India from 1st March 2020 to 31st May 2020. We find that the percentage reduction observed by satellites and surface monitoring are consistent with each other and follow the same trend of the Google mobility reductions. The comparison is discussed in Section 3.8.

5. Simultaneous meteorology conditions. The authors mentioned that meteorology conditions constant during recent years by citing some references. In this way, the natural emissions are

not the driver of the emission changes. Since this is the foundation of the whole analysis, I recommend a sub-session to clarify this point.

We use ERA-5 data to investigate the changes in meteorological parameters (temperature, wind speed and boundary layer height) between BAU and lockdown and analyze the differences in probability density functions. The meteorological changes have been discussed in a new Section 3.1.

Specific comments:

1. Page 2, line 47. I suggest using the term of large to replace larger in the phase of larger localised emissions.

Thank you for the suggestion. The necessary change has been made in the text.

2. Page 2, line 59. The description of "spatio-temporal similarity with ground-based measurements" is confusing. Do the authors indicate the satellite and ground measurements share the similar spatial and temporal resolution?

Sorry for the confusing statement. We have modified the statement to "Spatio-temporal coincidence with ground-based measurements".

3. Figure 4. I suggest adding a map to show the definition of the domain of Central, NWest, IGP and so on. It will be easier for readers to follow.

A map of India along with different regions (shown in different colors) along with in-situ measurement locations is shown in SUP Fig. 1.

Authors' responses to Referee #2. Reviewer's comments are in black text and authors' responses are in blue text.

**Anonymous Referee #2**

Many papers have recently appeared in the literature, including papers on India, documenting reductions linked to COVID-19 measures. A large fraction of those papers made use of TROPOMI and OMI observations. Therefore, I asked myself the question: what is new, and what have I learned? The authors write: "While recent studies have utilised either only satellite observations or only surface observations, this study goes further by adopting an integrated approach by combining both measurement types" Indeed, the paper contains interesting plots showing how surface and satellite analyses agree well, and document the relative reductions during LDN for the regions and cities in India. What I found also interesting is the analysis of the land use dependence and impact of changes in fire activity. The paper by Biswal et al. is a well written, easy to read and clear paper, with good English. The paper contains an extended set of relevant references. Because of the above I am in favour of publishing the paper after my comments have been taken into account.

We thank you for your helpful comments and your positive views on the scientific novelty of our study. Your detailed review has helped us to improve the manuscript significantly. In the revised manuscript, the analysis has been updated using 5 km  $\times$  5 km TROPOMI VCDtrop NO2. We have also used ERA-5 data to investigate the changes in met parameters (temperature, wind speed and boundary layer height) between BAU and lockdown. Section 3.8 linking the Google mobility change with NO2 change has been added. The limitations of the study have also been discussed in Section 3.9.

Weather variability normally has a large impact on the BAU/LDN ratios and can easily cause differences of the order of 20% in local differences between years. It is a bit of a pity that this aspect has not been explored by the authors. The paper is based on observations only, while models would need to be introduced to compensate for weather variability. Nevertheless, it would be good if the authors could provide a rough uncertainty estimate for the BAU/LDN relative changes due to the neglect of weather variability.

We agree with the reviewer's comment that weather variability can affect the BAU/LDN ratios of NO2. However, the effect will be a minimum under similar meteorological conditions. Recent studies have shown that meteorological conditions remained relatively consistent over recent years during the lockdown dates for India. Therefore, we believe that weather variability from year to year will have had less impact on NO2 changes. However, it is important to highlight the meteorological differences during the study period, as suggested by the reviewer, to assess the uncertainties associated with meteorological differences. We use ERA-5 data to investigate the changes in meteorological parameters (temperature, wind speed, and boundary layer height) between BAU and lockdown and analyze the differences in probability density functions. The meteorological changes have been discussed in a new Section 3.1.

Substantial differences are found between the OMI and TROPOMI products: can those be understood? Newer satellite datasets are available for OMI. Why have those not been used?

Regarding OMI, We used the version 4 OMI products but described them as version 3 because of confusing names (OMNO2d\_003). However, later it was clarified that OMNO2d\_003 is a version 4 product. We have explained it in the text.

For the comments "Newer satellite datasets are available for OMI. Why have those not been used?", we suggest that OMI, apart from TROPOMI, is the most relevant instrument to look at column changes in NO2. Since OMI, there has been the GOME-2A and GOME-2B instruments on MetOp-A (2006) and MetOp-B (2012). The resolution of these instruments is much coarser than OMI (e.g., 40 km  $\times$  80 km vs. 24 km  $\times$  13 km at nadir), making it much more difficult to see changes in NO2 source region. Therefore, OMI, alongside TROPOMI, is the most appropriate product to use.

For the comment "Substantial differences are found between the OMI and TROPOMI products: can those be understood?", it is beyond the scope of this study to investigate the core retrieval differences which are causing any OMI-TROPOMI differences. However, in our response on the page 3 (i.e. 1110) as well as in the revised manuscriptwe have outlined potential reasons for any differences between TROPOMI and OMI NO2, indicating the relevant literature for the reader to follow.

**It would be interesting to include also an analysis of the reductions of the major coal power plants, similar to the reductions observed for individual cities.**

As per the reviewer's suggestion, a separate section (3.5.3) has been added to analyze NO2 changes over coal thermal power plants (shown in Figure S5). From the analysis, it can be deduced that the thermal power plant regions have shown a reduction (with a maximum of 20-30% for IGP and south India), except the northeastern region. The reduction can be linked with the reported reduction in power demand and production during the lockdown period against the previous year (2019). The anomalous behavior of north eastern region can be due to the impact of forest fires, which is prevalent in this region during March-April.

Due to transport of pollution, background values will be largely caused by downstream NO2 from the cities and power plants. This is not discussed well enough, and influences the land-use dependent analysis.

We agree with the reviewer's view; in fact, this was one of the motivations behind estimating the changes over cropland and forestland (areas other than cities) to see how much the reduction over urban areas impacts the decrease in rural areas. We also agree that the absolute reduction will be larger in the downwind direction. However, this aspect has not been discussed in the present study because to pin-point the reductions in downstream from the cities or power plants, a comprehensive analysis including modeling will be required, which is beyond the scope of this work. Therefore, we have included this aspect as one of the limitations of this study and identified it as recommendations for a follow-up study.

**Detailed comments:**

Intro, page 3/4: when referring to "decreases" I would always like to know with respect to what: is it compared to climatology, to concentrations before the lockdown, or in comparison to 2019? I suggest this information is provided in most cases (now it is given for only some references).

We have revised the text to be clear and provided the information related to the reported changes concerning pre-lockdown or the same period in the previous years.

1110: "Moreover, as both instruments use similar retrieval schemes, their NO2 measurements should be comparable with a suitable degree of confidence?" I understand that there are also major differences between these retrievals. These could be highlighted more.

Both the KNMI TROPOMI and NASA OMI products use differential optical absorption spectroscopy (DOAS) to derive tropospheric column NO2. However, the inputs and methods between retrievals can differ. Key differences involve the representation of the stratospheric slant column, the assumed a priori profile, the treatment of aerosols/cloud and the surface albedo. For instance, the TROPOMI product uses data assimilation of independent stratospheric observations into a CTM to estimate the stratospheric slant column. On the other hand, NASA OMI product assumes that the stratospheric slant column dominates the total slant column over remote regions. Then model simulated small tropospheric component in remote areas is removed to estimate the stratospheric slant column that is then used to get tropospheric column over similar latitude bands. For a priori NO2 profile in the AMF calculation, TROPOMI uses TM5-MP, while the OMI product uses GEOS-Chem. The models will both have differences (input emissions, input meteorological files, different chemical schemes, etc.), which could also introduce differences in the NO2 products. The description of both schemes and their differences can be found in van Geffen et al. (2019) and Lamsal et al. (2021), respectively.

While it is beyond the scope of this study to outline the core retrieval differences which might be generating any differences between TROPOMI and OMI seen in this study, we have noted that the retrievals are similar. Therefore, we propose to update "Moreover, as both instruments use similar retrieval schemes, their NO2 measurements should be comparable with a suitable degree of confidence" with "Moreover, as both instruments use similar retrieval schemes (i.e., differential optical absorption spectroscopy, DOAS), their NO2 measurements should be comparable with a suitable degree of confidence (van Geffen et al., 2020; Wang et al., 2020). Any product differences are likely to be caused by inconsistent inputs/processing of the retrievals (e.g., derivation of the stratospheric slant column, the a priori tropospheric NO2 profile and the treatment of aerosols/clouds in the calculation of the air mass factor (Geffen et al., 2019; Lasmal et al., 2021))"

1147: The resolution of TROPOMI in 2020 is  $3.5 \times 5.5$  km. 1150: TROPOMI has a (2×) better resolution than  $10 \times 10$  km. Why was this resolution chosen?  $5 \times 5$  would be more appropriate.

Previously, we had used TROPOMI data having a resolution of  $0.1^{\circ} \times 0.1^{\circ}$  (~10 km × 10 km) obtained from the popular geospatial platform Google earth engine. Now, in the revised manuscript, we used TROPOMI VCDtrop NO2 over India at 3.5 km × 7 km resolution and regridded at a spatial resolution of  $0.05^{\circ} \times 0.05^{\circ}$  (~5 km × 5 km) based on the gridding methodology by Pope et al. (2018). All the analysis has been updated with the new dataset.

Sec 2.1.1, 2.1.2: I miss papers discussing the validation of the OMI and TROPOMI products.

As per the reviewer's suggestion, new references for the validation of the OMI and TROPOMI have been added in the text as well as in the reference section. The added references are:

van Geffen, J., Boersma, K. F., Eskes, H., Sneep, M., ter Linden, M., Zara, M., and Veefkind, J. P.: S5P TROPOMI NO2 slant column retrieval: method, stability, uncertainties and comparisons with OMI, Atmos. Meas. Tech., 13, 1315–1335, https://doi.org/10.5194/amt-13-1315-2020, 2020.

Chan, K. L., Wiegner, M., van Geffen, J., De Smedt, I., Alberti, C., Cheng, Z., Ye, S., and Wenig, M.: MAX-DOAS measurements of tropospheric NO2 and HCHO in Munich and the

comparison to OMI and TROPOMI satellite observations, Atmos. Meas. Tech., 13, 4499–4520, https://doi.org/10.5194/amt-13-4499-2020, 2020.

Wang, C.; Wang, T.; Wang, P.; Rakitin, V. Comparison and Validation of TROPOMI and OMI NO2 Observations over China. Atmosphere, 11, 636. 2020

Sec 2.1.1, 2.1.2: The authors suggest that the retrieval approaches are comparable. However, there are also major differences. Compared to NASA v3 there are two newer retrieval products available: NASA v4 and the QA4ECV products. Have the authors considered using those?

The NASA OMI tropospheric column data (OMNO2d\_003) is, in fact, version 4 data. We apologize for not explaining it (please see a response, 1110, on page 3). Earlier, we used time-averaged NO2 data from GIOVAANI platform. In the revised manuscript, we use the daily OMI NO2 30% Cloud-Screened Tropospheric Column L3 Global Gridded (Version 4) at a  $0.25^{\circ} \times 0.25^{\circ}$  (~25 km × 25 km) spatial grid from the NASA Goddard Earth Sciences Data and Information Services Center (GESDISC).

Sec. 2.1.3: Is there a classification of these stations available (e.g. traffic, urban background etc..)? Please comment. Can stations be compared, and is a correlation with the satellite data meaningful?

We agree with the reviewer that the classification of ground stations will clarify the pollutant concentrations and their variations. However, for India, there are no such classifications for CPCB station. Yet, most of the stations are in major urban areas.

Our analysis shows that surface observations are positively correlated with satellite observations. Although the correlation is low, because we use stations without classification, it gives us confidence that satellite observations capture the variability in the surface. To help the readers, we have added the explanation of low correlation and also stressed the need to make the station classification for scientific studies.

1188: "Details of the pollution mapping". Do you mean "population mapping"?

Thank you. It has been corrected.

Sec. 3.1. If I look at the satellite data it seems there is some impact of the fires, but the fire signal seems to be rather modest compared to the cities/power plants. For the fire anomalies I expect the differences to be even smaller. Why is it so important to exclude fire-influenced locations?

Forest fires are an important source of surface NO2 and VCDtrop NO2. In India, forest fires are prevalent as 36% of the country's forest cover is prone to frequent fires, out of which nearly 10% is extremely to very highly prone to fires and they peak during March to May.

Although it was difficult to remove the influence of long-range transport of fire plumes, we tried to reduce the uncertainty associated with forest fires by removing the fire and nearby grids. Please also see a response to reviewer 1.

1222: "we have considered the grids with zero fire anomaly to assess the changes in NO2 during the lockdown." Plumes from the fires may be transported over long distances, so blocking fire pixels does not mean the fire signals have been removed.

Yes, we agree with the reviewer. We have tried to minimize the impact changes in NO2 levels due to the forest fires, however, we have not removed the fire signals those are transported over long distances, as evident over the north-east region. This is one of the limitations of the work

that has been discussed in Section 3.9. Our study also demonstrated that decrease at the surface was masked by enhancement in NO2 due to the transport of the fire emissions.

258: "Background VCDtrop". How is it determined if a pixel is "background"?

Sorry for the confusion. Here "Background VCDtrop" means the rural NO2 and the statement has been modified.

1265: "biomass burning activities" From figure 1 it is not so clear if there was an increase or decrease in fire activity in the West. Could you comment on this?

Although the variations are not easily noticeable (due to resolution) in the West, there are some scattered grids where the fire activity is shown to be increased.

Figure 1. Fire anomalies in the West

Sec 3.4: "The change observed over different land use" This title is a bit unclear, please reformulate

Section 3.4 is now 3.5 and the title has been modified to "Changes in NO2 over different landuse types"

Sec 3.4.1: Land use changes occur over short distances. In contrast, the transport of (elevated) NO2 can happen over hundreds of kilometres. Therefore, the emissions from urban areas or coal plants will enhance concentrations over crop/forest grid cells, which blurs the interpretation. Also, differences in spatial resolution between OMI and TROPOMI may have similar effects. As such, there is a bit of a danger to over-interpret the results of e.g. Fig. 4.

Yes, we agree with the viewpoint of the reviewer. We have put a significant efforts to assign the corresponding land-use type corresponding to the OMI and TROPOMI data grid. We also agree that the pollution transport from urban and TPPs to crop/forest areas may affect the observed changes. However, this is one of the aims of this study to estimate the lockdowninduced changes in NO2 levels over crop/forest areas where the anthropogenic emissions are minimal.

Sec. 3.4.2: I would suggest not to repeat numbers from the table inside the text. Better to highlight a few.

Thank you for the suggestion. We have now removed the numbers and only the key messages are now kept in the text.

1339: "The actual reduction in VCDtrop NO2 is greater for the larger urban area". But what is the reason? Is it related to different reductions in those cities or to differences in the relative contribution of the background not caused by emissions inside the city?

The more significant reduction for the larger urban areas is mainly due to a decrease in local emission sources. Considering NO2 emission dominated by the transport sector, we investigated the change in mobility for small to large cities. The mobility reduction over larger cities (larger population) is higher than smaller ones suggesting the different reduction in cities.

*Figure 2. Scatter of the mobility change with the population*

Sec 3.5, fig. 6: Without knowledge of the station classification or location relative to the sources these plots are not so meaningful. I would suggest to remove them, or put them in the supplement. Combining road-side and urban background stations in the plots will strongly reduce correlations.

We agree, but we prefer to keep this plot in the manuscript as it demonstrates the need to make the station classification clearer for scientific studies. It provides support for our recommendation. We have added an explanation of low correlation in Section 3.6 because of the unavailability of the station classification and added as one of the limitations of this study.

Fig. 7: Ahmedabad is a clear outlier. Can the difference between surface and satellite observations be understood, for instance with information on the locations of the surface sites?

Ahmedabad is surrounded by industries, but we did not include the site from the industrial area. However, non-industrial sites show a reduction of  $\sim$ 70%, most probably because of the decline in traffic emissions along with the industrial emissions.

1411: "adding a constant". This is not a logical thing to do in case of log-log plots, because the impact will be very non-linear. But the vertical scale in Figure 8c is linear? Please re-write or remove the explanation.

The addition of a constant for the log transformation of the data having zero or negative values is a standard practice in data analysis (Ekwaru et al., 2018. St-Pierre et al., 2018). The purpose

of Figure 8c is to show the relationship between reduction and population. The scale in Figure 8c is not linear as the gap between 0-50 is smaller than the gap between 150 to 200.

Ekwaru, J. P. and Veugelers, P. J.: The Overlooked Importance of Constants Added in Log Transformation of Independent Variables with Zero Values: A Proposed Approach for Determining an Optimal Constant, Stat. Biopharm. Res., 10(1), 26–29, https://doi.org/10.1080/19466315.2017.1369900, 2018.

St-Pierre, A. P., Shikon, V. and Schneider, D. C.: Count data in biology-Data transformation or model reformation?, Ecol. Evol., 8(6), 3077–3085, https://doi.org/10.1002/ece3.3807, 2018.

1 415: It is not easy for me to judge from the figure panels if the relation follows a power law. And what does a factor 2.0 for OMI tell me?

The linear best fit lines show the log-log relationship between VCDtrop NO2 (Y) and population density (X) given by equation  $y=\beta.x+c$ , where  $y=\log(Y)$ ,  $x=\log(X)$  and  $c=\log(C)$ . Therefore, the equation can be written as

 $log(Y) = \beta .log(X) + log(C)$ , i.e., can be written in power-law  $Y=C.X^{\beta}$  where  $\beta$  is the slope of the line and defines the relationship between VCDtrop NO2 and population and C is the average VCDtrop NO2 for the grid having unit population per hectare.

So, for a 10-fold change in population density, the VCDtrop NO2 will change by a factor of  $10^{0.28} = 1.9$ . A similar approach has been discussed in (Lamsal et al., 2013).